# A Fourier Perspective on Model Robustness in Computer Vision

**Dong Yin**[*]
Department of EECS
UC Berkeley
Berkeley, CA 94720
dongyin@berkeley.edu

**Raphael Gontijo Lopes**[†]
Google Research, Brain team
Mountain View, CA 94043
iraphael@google.com

**Jonathon Shlens**
Google Research, Brain team
Mountain View, CA 94043
shlens@google.com

**Ekin D. Cubuk**
Google Research, Brain team
Mountain View, CA 94043
cubuk@google.com

**Justin Gilmer**
Google Research, Brain team
Mountain View, CA 94043
gilmer@google.com

## Abstract

Achieving robustness to distributional shift is a longstanding and challenging goal of computer vision. Data augmentation is a commonly used approach for improving robustness, however robustness gains are typically not uniform across corruption types. Indeed increasing performance in the presence of random noise is often met with reduced performance on other corruptions such as contrast change. Understanding when and why these sorts of trade-offs occur is a crucial step towards mitigating them. Towards this end, we investigate recently observed trade-offs caused by Gaussian data augmentation and adversarial training. We find that both methods improve robustness to corruptions that are concentrated in the high frequency domain while reducing robustness to corruptions that are concentrated in the low frequency domain. This suggests that one way to mitigate these trade-offs via data augmentation is to use a more diverse set of augmentations. Towards this end we observe that AutoAugment [6], a recently proposed data augmentation policy optimized for clean accuracy, achieves state-of-the-art robustness on the CIFAR-10-C [17] benchmark.

## 1 Introduction

Although many deep learning computer vision models achieve remarkable performance on many standard i.i.d benchmarks, these models lack the robustness of the human vision system when the train and test distributions differ [24]. For example, it has been observed that commonly occurring image corruptions, such as random noise, contrast change, and blurring, can lead to significant performance degradation [8, 3]. Improving distributional robustness is an important step towards safely deploying models in complex, real-world settings.

Data augmentation is a natural and sometimes effective approach to learning robust models. Examples of data augmentation include adversarial training [14], applying image transformations to the training data, such as flipping, cropping, adding random noise, and even stylized image transformation [11].

However, data augmentation rarely improves robustness across all corruption types. Performance gains on some corruptions may be met with dramatic reduction on others. As an example, in [10] it

---

[*]Work done while internship at Google Research, Brain team.
[†]Work done as a member of the Google AI Residency program g.co/airesidency.

was observed that Gaussian data augmentation and adversarial training improve robustness to noise and blurring corruptions on the CIFAR-10-C and ImageNet-C common corruption benchmarks [17], while significantly degrading performance on the fog and contrast corruptions. This begs a natural question

*What is different about the corruptions for which augmentation strategies improve performance vs. those which performance is degraded?*

Understanding these tensions and why they occur is an important first step towards designing robust models. Our operating hypothesis is that the frequency information of these different corruptions offers an explanation of many of these observed trade-offs. Through extensive experiments involving perturbations in the Fourier domain, we demonstrate that these two augmentation procedures bias the model towards utilizing low frequency information in the input. This low frequency bias results in improved robustness to corruptions which are more high frequency in nature while degrading performance on corruptions which are low frequency.

Our analysis suggests that more diverse data augmentation procedures could be leveraged to mitigate these observed trade-offs, and indeed this appears to be true. In particular we demonstrate that the recently proposed AutoAugment data augmentation policy [6] achieves state-of-the-art results on the CIFAR-10-C benchmark. In addition, a follow-up work has utilized AutoAugment in a way to achieve state-of-the-art results on ImageNet-C [1].

Some of our observations could be of interest to research on security. For example, we observe perturbations in the Fourier domain which when applied to images cause model error rates to exceed 90% on ImageNet while preserving the semantics of the image. These qualify as simple, single query[3] black box attacks that satisfy the content preserving threat model [13]. This observation was also made in concurrent work [26].

Finally, we extend our frequency analysis to obtain a better understanding of worst-case perturbations of the input. In particular adversarial perturbations of a naturally trained model are more high-frequency in nature while adversarial training encourages these perturbations to become more concentrated in the low frequency domain.

## 2 Preliminaries

We denote the $\ell_2$ norm of vectors (and in general, tensors) by $\|\cdot\|$. For a vector $x \in \mathbb{R}^d$, we denote its entries by $x[i]$, $i \in \{0, \ldots, d-1\}$, and for a matrix $X \in \mathbb{R}^{d_1 \times d_2}$, we denote its entries by $X[i,j]$, $i \in \{0, \ldots, d_1 - 1\}$, $j \in \{0, \ldots, d_2 - 1\}$. We omit the dimension of image channels, and denote them by matrices $X \in \mathbb{R}^{d_1 \times d_2}$. We denote by $\mathcal{F} : \mathbb{R}^{d_1 \times d_2} \to \mathbb{C}^{d_1 \times d_2}$ the 2D discrete Fourier transform (DFT) and by $\mathcal{F}^{-1}$ the inverse DFT. When we visualize the Fourier spectrum, we always shift the low frequency components to the center of the spectrum.

We define high pass filtering with bandwidth $B$ as the operation that sets all the frequency components outside of a centered square with width $B$ in the Fourier spectrum with highest frequency in the center to zero, and then applies inverse DFT. The low pass filtering operation is defined similarly with the difference that the centered square is applied to the Fourier spectrum with low frequency shifted to the center.

We assume that the pixels take values in range $[0,1]$. In all of our experiments with data augmentation we always clip the pixel values to $[0,1]$. We define Gaussian data augmentation with parameter $\sigma$ as the following operation: In each iteration, we add i.i.d. Gaussian noise $\mathcal{N}(0, \tilde{\sigma}^2)$ to every pixel in all the images in the training batch, where $\tilde{\sigma}$ is chosen uniformly at random from $[0, \sigma]$. For our experiments on CIFAR-10, we use the Wide ResNet-28-10 architecture [27], and for our experiment on ImageNet, we use the ResNet-50 architecture [16]. When we use Gaussin data augmentation, we choose parameter $\sigma = 0.1$ for CIFAR-10 and $\sigma = 0.4$ for ImageNet. All experiments use flip and crop during training.

**Fourier heat map** We will investigate the sensitivity of models to high and low frequency corruptions via a perturbation analysis in the Fourier domain. Let $U_{i,j} \in \mathbb{R}^{d_1 \times d_2}$ be a real-valued matrix such that $\|U_{i,j}\| = 1$, and $\mathcal{F}(U_{i,j})$ only has up to two non-zero elements located at $(i,j)$ and the its symmetric coordinate with respect to the image center; we call these matrices the 2D *Fourier basis* matrices [4].

Given a model and a validation image $X$, we can generate a perturbed image with Fourier basis noise. More specifically, we can compute $\widetilde{X}_{i,j} = X + rvU_{i,j}$, where $r$ is chosen uniformly at random from $\{-1, 1\}$, and $v > 0$ is the norm of the perturbation. For multi-channel images, we perturb every channel independently. We can then evaluate the models under Fourier basis noise and visualize how the test error changes as a function of $(i, j)$, and we call these results the Fourier heat map of a model. We are also interested in understanding how the outputs of the models' intermediate layers change when we perturb the images using a specific Fourier basis, and these results are relegated to the Appendix.

## 3 The robustness problem

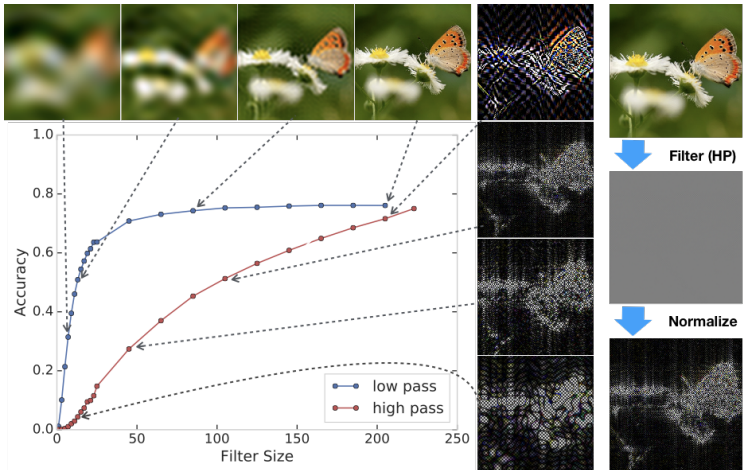

**Figure 1:** Models can achieve high accuracy using information from the input that would be unrecognizable to humans. Shown above are models trained and tested with aggressive high and low pass filtering applied to the inputs. With aggressive low-pass filtering, the model is still above 30% on ImageNet when the images appear to be simple globs of color. In the case of high-pass (HP) filtering, models can achieve above 50% accuracy using features in the input that are nearly invisible to humans. As shown on the right hand side, the high pass filtered images needed be normalized in order to properly visualize the high frequency features (the method that we use to visualize the high pass filtered images is provided in the appendix).

How is it possible that models achieve such high performance in the standard settings where the training and test data are i.i.d., while performing so poorly in the presence of even subtle distributional shift? There has been substantial prior work towards obtaining a better understanding of the *robustness problem*. While this problem is far from being completely understood, perhaps the simplest explanation is that models lack robustness to distributional shift simply because there is no reason for them to be robust [20, 11, 18]. In naturally occurring data there are many correlations between the input and target that models can utilize to generalize well. However, utilizing such sufficient statistics will lead to dramatic reduction in model performance should these same statistics become corrupted at test time.

As a simple example of this principle, consider Figure 8 in [19]. The authors experimented with training models on a "cheating" variant of MNIST, where the target label is encoded by the location of a single pixel. Models tested on images with this "cheating" pixel removed would perform poorly. This is an unfortunate setting where Occam's razor can fail. The simplest explanation of the data may generalize well in perfect settings where the training and test data are i.i.d., but fail to generalize *robustly*. Although this example is artificial, it is clear that model brittleness is tied to latching onto non-robust statistics in naturally occurring data.

As a more realistic example, consider the recently proposed *texture hypothesis* [11]. Models trained on natural image data can obtain high classification performance relying on local statistics that are correlated with texture. However, texture-like information can become easily distorted due to naturally occurring corruptions caused by weather or digital artifacts, leading to poor robustness.

In the image domain, there is a plethora of correlations between the input and target. Simple statistics such as colors, local textures, shapes, even unintuitive high frequency patterns can all be leveraged in a way to achieve remarkable i.i.d generalization. To demonstrate, we experimented with training

and testing of ImageNet models when severe filtering is performed on the input in the frequency domain. While modest filtering has been used for model compression [9], we experiment with extreme filtering in order to test the limits of model generalization. The results are shown in Figure 1. When low-frequency filtering is applied, models can achieve over 30% test accuracy even when the image appears to be simple globs of color. Even more striking, models achieve 50% accuracy in the presence of the severe high frequency filtering, using high frequency features which are nearly invisible to humans. In order to even visualize these high frequency features, we had normalize pixel statistics to have unit variance. Given that these types features are useful for generalization, it is not so surprising that models leverage these non-robust statistics.

It seems likely that these invisible high frequency features are related to the experiments of [18], which show that certain imperceptibly perturbed images contain features which are useful for generalization. We discuss these connections more in Section 4.4.

## 4   Trade-off and correlation between corruptions: a Fourier perspective

The previous section demonstrated that both high and low frequency features are useful for classification. A natural hypothesis is that data augmentation may bias the model towards utilizing different kinds of features in classification. What types of features models utilize will ultimately determine the robustness at test time. Here we adopt a Fourier perspective to study the trade-off and correlation between corruptions when we apply several data augmentation methods.

### 4.1   Gaussian data augmentation and adversarial training bias models towards low frequency information

Ford et al. [10] investigated the robustness of three models on CIFAR-10-C: a naturally trained model, a model trained by Gaussian data augmentation, and an adversarially trained model. It was observed that Gaussian data augmentation and adversarial training improve robustness to all noise and many of the blurring corruptions, while degrading robustness to fog and contrast. For example adversarial training degrades performance on the most severe contrast corruption from 85.66% to 55.29%. Similar results were reported on ImageNet-C.

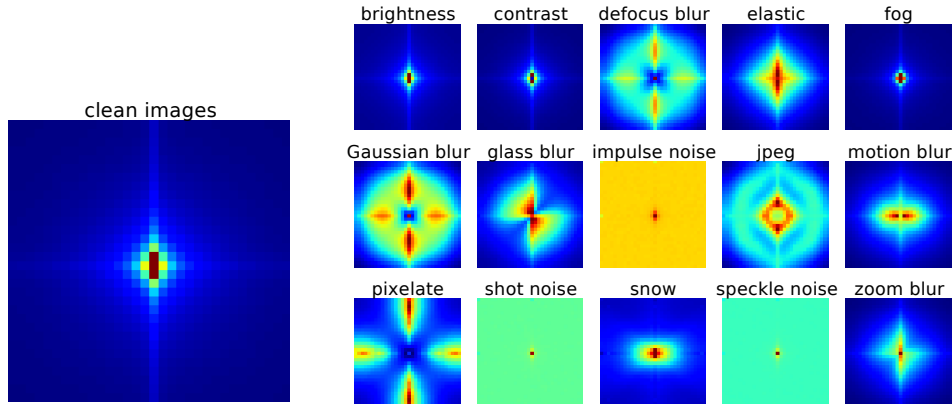

**Figure 2:** Left: Fourier spectrum of natural images; we estimate $\mathbb{E}[|\mathcal{F}(X)[i,j]|]$ by averaging all the CIFAR-10 validation images. Right: Fourier spectrum of the corruptions in CIFAR-10-C at severity 3. For each corruption, we estimate $\mathbb{E}[|\mathcal{F}(C(X)-X)[i,j]|]$ by averaging over all the validation images. Additive noise has relatively high concentrations in high frequencies while some corruptions such as fog and contrast are concentrated in low frequencies.

We hypothesize that some of these trade-offs can be explained by the Fourier statistics of different corruptions. Denote a (possibly randomized) corruption function by $C : \mathbb{R}^{d_1 \times d_2} \to \mathbb{R}^{d_1 \times d_2}$. In Figure 2 we visualize the Fourier statistics of natural images as well as the average delta of the common corruptions. Natural images have higher concentrations in low frequencies, thus when we refer to a "high" or "low" frequency corruption we will always use this term on a relative scale. Gaussian noise is uniformly distributed across the Fourier frequencies and thus has much higher frequency statistics relative to natural images. Many of the blurring corruptions remove or change the high frequency content of images. As a result $C(X) - X$ will have a higher fraction of high frequency

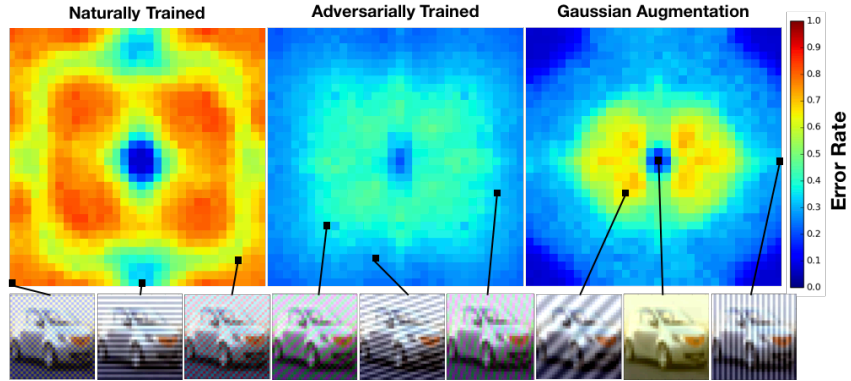

**Figure 3:** Model sensitivity to additive noise aligned with different Fourier basis vectors on CIFAR-10. We fix the additive noise to have $\ell_2$ norm 4 and evaluate three models: a naturally trained model, an adversarially trained model, and a model trained with Gaussian data augmentation. Error rates are averaged over 1000 randomly sampled images from the test set. In the bottom row we show images perturbed with noise along the corresponding Fourier basis vector. The naturally trained model is highly sensitive to additive noise in all but the lowest frequencies. Both adversarial training and Gaussian data augmentation dramatically improve robustness in the higher frequencies while sacrificing the robustness of the naturally trained model in the lowest frequencies (i.e. in both models, blue area in the middle is smaller compared to that of the naturally trained model).

energy. For corruptions such as contrast and fog, the energy of the corruption is concentrated more on low frequency components.

The observed differences in the Fourier statistics suggests an explanation for why the two augmentation methods improve performance in additive noise but not fog and contrast — the two augmentation methods encourage the model to become invariant to high frequency information while relying more on low frequency information. We investigate this hypothesis via several perturbation analyses of the three models in question. First, we test model sensitivity to perturbations along each Fourier basis vector. Results on CIFAR-10 are shown in Figure 3. The difference between the three models is striking. The naturally trained model is highly sensitive to additive perturbations in all but the lowest frequencies, while Gaussian data augmentation and adversarial training both dramatically improve robustness in the higher frequencies. For the models trained with data augmentation, we see a subtle but distinct lack of robustness at the lowest frequencies (relative to the naturally trained model). Figure 4 shows similar results for three different models on ImageNet. Similar to CIFAR-10, Gaussian data augmentation improves robustness to high frequency perturbations while reducing performance on low frequency perturbations.

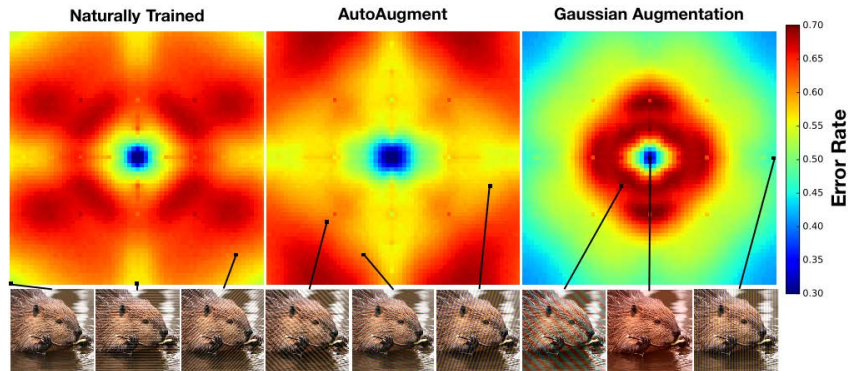

**Figure 4:** Model sensitivity to additive noise aligned with different Fourier basis vectors on ImageNet validation images. We fix the basis vectors to have $\ell_2$ norm 15.7. Error rates are averaged over the entire ImageNet validation set. We present the $63 \times 63$ square centered at the lowest frequency in the Fourier domain. Again, the naturally trained model is highly sensitive to additive noise in all but the lowest frequencies. On the other hand, Gaussian data augmentation improves robustness in the higher frequencies while sacrificing the robustness to low frequency perturbations. For AutoAugment, we observe that its Fourier heat map has the largest blue/yellow area around the center, indicating that AutoAugment is relatively robust to low to mid frequency corruptions.

To test this further, we added noise with fixed $\ell_2$ norm but different frequency bandwidths centered at the origin. We consider two settings, one where the origin is centered at the lowest frequency and one where the origin is centered at the highest frequency. As shown in Figure 5, for a low frequency centered bandwidth of size 3, the naturally trained model has less than half the error rate of the other two models. For high frequency bandwidth, the models trained with data augmentation dramatically outperform the naturally trained model.

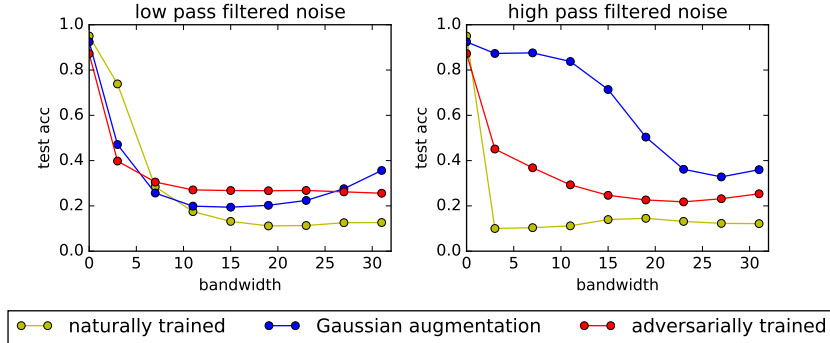

**Figure 5:** Robustness of models under additive noise with fixed norm and different frequency distribution. For each channel in each CIFAR-10 test image, we sample i.i.d Gaussian noise, apply a low/high pass filter, and normalize the filtered noise to have $\ell_2$ norm 8, before applying to the image. We vary the bandwidth of the low/high pass filter and generate the two plots. The naturally trained model is more robust to the low frequency noise with bandwidth 3, while Gaussian data augmentation and adversarial training make the model more robust to high frequency noise.

This is consistent with the hypothesis that the models trained with the noise augmentation are biased towards low frequency information. As a final test, we analyzed the performance of models with a low/high pass filter applied to the input (we call the low/high pass filters the *front end* of the model). Consistent with prior experiments we find that applying a low pass front-end degrades performance on fog and contrast while improving performance on additive noise and blurring. If we instead further bias the model towards high frequency information we observe the opposite effect. Applying a high-pass front end degrades performance on all corruptions (as well as clean test error), but performance degradation is more severe on the high frequency corruptions. These experiments again confirm our hypothesis about the robustness properties of models with a high (or low) frequency bias.

To better quantify the relationship between frequency and robustness for various models we measure the ratio of energy in the high and low frequency domain. For each corruption $C$, we apply high pass filtering with bandwidth 27 (denote this operation by $H(\cdot)$) on the delta of the corruption, i.e., $C(X) - X$. We use $\frac{\|H(C(X)-X)\|^2}{\|C(X)-X\|^2}$ as a metric of the fraction of high frequency energy in the corruption. For each corruption, we average this quantity over all the validation images and all 5 severities. We evaluate 6 models on CIFAR-10-C, each trained differently — natural training, Gaussian data augmentation, adversarial training, trained with a low pass filter front end (bandwidth 15), trained with a high pass filter front end (bandwidth 31), and trained with AutoAugment (see a more detailed discussion on AutoAugment in Section 4.3). Results are shown in Figure 6. Models with a low frequency bias perform better on the high frequency corruptions. The model trained with a high pass filter has a forced high frequency bias. While this model performs relatively poorly on even natural data, it is clear that high frequency corruptions degrade performance more than the low frequency corruptions. Full results, including those on ImageNet, can be found in the appendix.

## 4.2 Does low frequency data augmentation improve robustness to low frequency corruptions?

While Figure 6 shows a clear relationship between frequency and robustness gains of several data augmentation strategies, the Fourier perspective is not predictive in all situations of transfer between data augmentation and robustness.

We experimented with applying additive noise that matches the statistics of the fog corruption in the frequency domain. We define "fog noise" to be the additive noise distribution $\sum_{i,j} \mathcal{N}(0, \sigma_{i,j}^2) U_{i,j}$

where the $\sigma_{i,j}$ are chosen to match the typical norm of the fog corruption on basis vector $U_{i,j}$ as

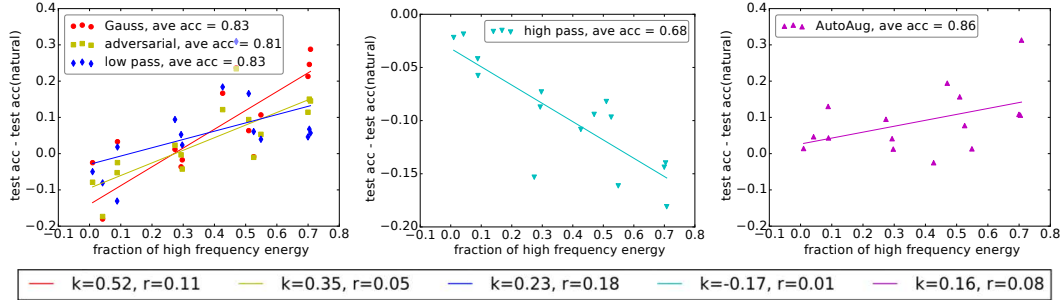

**Figure 6:** Relationship between test accuracy and fraction of high frequency energy of the CIFAR-10-C corruptions. Each scatter point in the plot represents the evaluation result of a particular model on a particular corruption type. The x-axis represents the fraction of high frequency energy of the corruption type, and the y-axis represents change in test accuracy compared to a naturally trained model. Overall, Gaussian data augmentation, adversarial training, and adding low pass filter improve robustness to high frequency corruptions, and degrade robustness to low frequency corruptions. Applying a high pass filter front end yields a more significant accuracy drop on high frequency corruptions compared to low frequency corruptions. AutoAugment improves robustness on nearly all corruptions, and achieves the best overall performance. The legend at the bottom shows the slope ($k$) and residual ($r$) of each fitted line.

shown in Figure 2. In particular, the marginal statistics of fog noise are identical to the fog corruption in the Fourier domain. However, data augmentation on fog noise *degrades* performance on the fog corruption (Table 1). This occurs despite the fact that the resulting model yields improved robustness to perturbations along the low frequency vectors (see the Fourier heat maps in the appendix).

| fog severity | 1 | 2 | 3 | 4 | 5 |
|---|---|---|---|---|---|
| naturally trained | 0.9606 | 0.9484 | 0.9395 | 0.9072 | 0.7429 |
| fog noise augmentation | 0.9090 | 0.8726 | 0.8120 | 0.7175 | 0.4626 |

**Table 1:** Training with fog noise hurts performance on fog corruption.

We hypothesize that the story is more complicated for low frequency corruptions because of an asymmetry between high and low frequency information in natural images. Given that natural images are concentrated more in low frequencies, a model can more easily learn to "ignore" high frequency information rather than low frequency information. Indeed as shown in Figure 1, model performance drops off far more rapidly when low frequency information is removed than high.

### 4.3 More varied data augmentation offers more general robustness

The trade-offs between low and high frequency corruptions for Gaussian data augmentation and adversarial training lead to the natural question of how to achieve robustness to a more diverse set of corruptions. One intuitive solution is to train on a variety of data augmentation strategies. Towards this end, we investigated the learned augmentation policy AutoAugment [6]. AutoAugment applies a learned mixture of image transformations during training and achieves the state-of-the-art performance on CIFAR-10 and ImageNet. In all of our experiments with AutoAugment, we remove the brightness and contrast sub-policies as they explicitly appear in the common corruption benchmarks. [4] Despite the fact that this policy was tuned specifically for clean test accuracy, we found that it also dramatically improves robustness on CIFAR-10-C. Here, we demonstrate part of the results in Table 2, and the full results can be found in the appendix. In the third plot in Figure 6, we also visualize the performance of AutoAugment on CIFAR-10-C.

More specifically, on CIFAR-10-C, we compare the robustness of the naturally trained model, Gaussian data augmentation, adversarially trained model, and AutoAugment. We observe that among the four models, AutoAugment achieves the best average corruption test accuracy of 86%. Using the mean corruption error (mCE) metric proposed in [17] with the naturally trained model being the baseline (see a formal definition of mCE in the appendix), we observe that AutoAugment achieves the best mCE of 64, and in comparison, Gaussian data augmentation and adversarial training achieve mCE of 98 and 108, respectively. In addition, as we can see, AutoAugment improves robustness on all but one of the corruptions, compared to the naturally trained model.

`https://github.com/tensorflow/models/tree/master/research/autoaugment`.

| | | | noise | | | blur | | | | | weather | | | digital | | | |
|---|---|---|---|---|---|---|---|---|---|---|---|---|---|---|---|---|---|
| model | acc | mCE | speckle | shot | impulse | defocus | Gauss | glass | motion | zoom | snow | fog | bright | contrast | elastic | pixel | jpeg |
| natural | 77 | 100 | 70 | 68 | 54 | 85 | 73 | 57 | 81 | 80 | 85 | 90 | 95 | 82 | 86 | 73 | 80 |
| Gauss | 83 | 98 | **92** | **92** | 83 | 84 | 79 | **80** | 77 | 82 | 88 | 72 | 92 | 57 | 84 | **90** | **91** |
| adversarial | 81 | 108 | 82 | 83 | 69 | 84 | 82 | **80** | 80 | 83 | 83 | 73 | 87 | 77 | 82 | 85 | 85 |
| Auto | **86** | **64** | 81 | 78 | **86** | **92** | **88** | 76 | **85** | **90** | **89** | **95** | **96** | **95** | **87** | 71 | 81 |

**Table 2:** Comprison between naturally trained model (natural), Gaussian data augmentation (Gauss), adversarially trained model (adversarial), and AutoAugment (Auto) on CIFAR-10-C. We remove all corruptions that appear in this benchmark from the AutoAugment policy. All numbers are in percentage. The first column shows the average top1 test accuracy on all the corruptions; the second column shows the mCE; the rest of the columns show the average test accuracy over the 5 severities for each corruption. We observe that AutoAugment achieves the best average test accuracy and the best mCE. In most of the blurring and all of the weather corruptions, AutoAugment achieves the best performance among the four models.

As for the ImageNet-C benchmark, instead of using the compressed ImageNet-C images provided in [17], we evaluate the models on corruptions applied in memory, [5] and observe that AutoAugment also achieves the highest average corruption test accuracy. The full results can be found in the appendix. As for the compressed ImageNet-C images, we note that a follow-up work has utilized AutoAugment in a way to achieve state-of-the-art results [1].

### 4.4 Adversarial examples are not strictly a high frequency phenomenon

Adversarial perturbations remain a popular topic of study in the machine learning community. A common hypothesis is that adversarial perturbations lie primarily in the high frequency domain. In fact, several (unsuccessful) defenses have been proposed motivated specifically by this hypothesis. Under the assumption that compression removes high frequency information, JPEG compression has been proposed several times [21, 2, 7] as a method for improving robustness to small perturbations. Studying the statistics of adversarially generated perturbations is not a well defined problem because these statistics will ultimately depend on how the adversary constructs the perturbation. This difficulty has led to many false claims of methods for detecting adversarial perturbations [5]. Thus the analysis presented here is to better understand common hypothesis about adversarial perturbations, rather than actually detect all possible perturbations.

For several models we use PGD to construct adversarial perturbations for every image in the test set. We then analyze the delta between the clean and perturbed images and project these deltas into the Fourier domain. By aggregating across the successful attack images, we obtain an understanding of the frequency properties of the constructed adversarial perturbations. The results are shown in Figure 7.

For the naturally trained model, the measured adversarial perturbations do indeed show higher concentrations in the high frequency domain (relative to the statistics of natural images). However, for the adversarially trained model this is no longer the case. The deltas for the adversarially trained model resemble that of natural data. Our analysis provides some additional understanding on a number of observations in prior works on adversarial examples. First, while adversarial perturbations for the naturally trained model do indeed show higher concentrations in the high frequency domain, this does not mean that removing high frequency information from the input results in a robust model. Indeed as shown in Figure 3, the naturally trained model is not worst-case or even average-case robust on any frequency (except perhaps the extreme low frequencies). Thus, we should expect that if we adversarially searched for errors in the low frequency domain, we will find them easily. This explains why JPEG compression, or any other method based on specifically removing high frequency content, should not be expected to be robust to worst-case perturbations.

Second, the fact that adversarial training biases these perturbations towards the lower frequencies suggests an intriguing connection between adversarial training and the DeepViz [23] method for feature visualization. In particular, optimizing the input in the low frequency domain is one of the strategies utilized by DeepViz to bias the optimization in the image space towards semantically meaningful directions. Perhaps the reason adversarially trained models have semantically meaningful gradients [25] is because gradients are biased towards low frequencies in a similar manner as utilized in DeepViz.

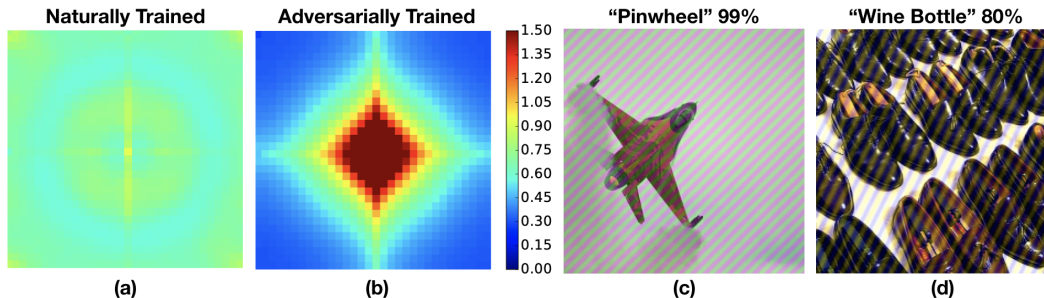

**Figure 7:** (a) and (b): Fourier spectrum of adversarial perturbations. For any image $X$, we run the PGD attack [22] to generate an adversarial example $C(X)$. We estimate the Fourier spectrum of the adversarial perturbation, i.e., $\mathbb{E}[|\mathcal{F}(C(X) - X)[i, j]|]$, where the expectation is taken over the perturbed images which are incorrectly classified. (a) naturally trained; (b) adversarially trained. The adversarial perturbations for the naturally trained model are uniformly distributed across frequency components. In comparison, adversarial training biases these perturbations towards the lower frequencies. (c) and (d): Adding Fourier basis vectors with large norm to images is a simple method for generating content-preserving black box adversarial examples.

As a final note, we observe that adding certain Fourier basis vectors with large norm (24 for ImageNet) degrades test accuracy to less than 10% while preserving the semantics of the image. Two examples of the perturbed images are shown in Figure 7. If additional model queries are allowed, subtler perturbations will suffice — the perturbations used in Figure 4 can drop accuracies to less than 30%. Thus, these Fourier basis corruptions can be considered as content-preserving black box attacks, and could be of interest to research on security. Fourier heat maps with larger perturbations are included in the appendix.

## 5 Conclusions and future work

We obtained a better understanding of trade-offs observed in recent robustness work in the image domain. By investigating common corruptions and model performance in the frequency domain we establish connections between frequency of a corruption and model performance under data augmentation. This connection is strongest for high frequency corruptions, where Gaussian data augmentation and adversarial training bias the model towards low frequency information in the input. This results in improved robustness to corruptions with higher concentrations in the high frequency domain at the cost of reduced robustness to low frequency corruptions and clean test error.

Solving the robustness problem via data augmentation alone feels quite challenging given the trade-offs we commonly observe. Naively augmenting on different corruptions often will not transfer well to held out corruptions [12]. However, the impressive robustness of AutoAugment gives us hope that data augmentation done properly can play a crucial role in mitigating the robustness problem.

Care must be taken though when utilizing data augmentation for robustness to not overfit to the validation set of held out corruptions. The goal is to learn *domain invariant features* rather than simply become robust to a specific set of corruptions. The fact that AutoAugment was tuned specifically for clean test error, and transfers well even after removing the contrast and brightness parts of the policy (as these corruptions appear in the benchmark) gives us hope that this is a step towards more useful domain invariant features. The robustness problem is certainly far from solved, and our Fourier analysis shows that the AutoAugment model is not strictly more robust than the baseline — there are frequencies for which robustness is degraded rather than improved. Because of this, we anticipate that robustness benchmarks will need to evolve over time as progress is made. These trade-offs are to be expected and researchers should actively search for new blindspots induced by the methods they introduce. As we grow in our understanding of these trade-offs we can design better benchmarks to obtain a more comprehensive perspective on model robustness.

While data augmentation is perhaps the most effective method we currently have for the robustness problem, it seems unlikely that data augmentation *alone* will provide a complete solution. Towards that end it will be important to develop orthogonal methods — e.g. architectures with better inductive biases or loss functions which when combined with data augmentation encourage extrapolation rather than interpolation.

**Acknowledgments**

We would like to thank Nicolas Ford and Norman Mu for helpful discussions.

## Footnotes

[3]In contrast, methods for generating small adversarial perturbations require 1000's of queries [15].

[4]Our experiment is based on the open source implementation of AutoAugment at

[5]The dataset of images with corruptions in memory can be found at `https://github.com/tensorflow/datasets/blob/master/tensorflow_datasets/image/imagenet2012_corrupted.py`.

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
