[Supplementary Material]

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

# Appendix

## A Comparison of model robustness on all the corruptions in CIFAR-10-C and ImageNet-C

We first define *mCE*, a quantity that we use to measure the robustness improvement of the models compared to a baseline model. Consider a total of $K$ corruptions, each with $S$ severities. Let $f$ be a model, and $E_{k,s}(f)$ be the model's test error under the $k$-th corruption in the benchmark with severity $s$, $k = 1, \ldots, K$, $s = 1, \ldots, S$. Let $f_0$ be the baseline model. We define mCE as the following quantity:

$$\text{mCE} = \frac{1}{K} \sum_{k=1}^{K} \frac{\sum_{s=1}^{S} E_{k,s}(f)}{\sum_{s=1}^{S} E_{k,s}(f_0)}.$$

For our CIFAR-10-C results in Table 3, we use the naturally trained WideResNet model as the baseline model. We present the full test accuracy results on CIFAR-10-C and ImageNet-C in Tables 3 and 4, respectively.

| | natural | Gauss | adversarial | low pass | high pass | AutoAugment | all-but-one |
|---|---|---|---|---|---|---|---|
| clean images | 0.9626 | 0.9369 | 0.8725 | 0.9235 | 0.9378 | **0.9693** | 0.9546 |
| brightness | 0.9493 | 0.9244 | 0.8705 | 0.8996 | 0.9275 | **0.9635** | 0.9407 |
| contrast | 0.8225 | 0.5703 | 0.7700 | 0.6917 | 0.7806 | **0.9526** | 0.9015 |
| defocus blur | 0.8456 | 0.8371 | 0.8355 | 0.9063 | 0.7489 | **0.9229** | 0.9495 |
| elastic transform | 0.8600 | 0.8429 | 0.8175 | **0.8838** | 0.7870 | 0.8726 | 0.9221 |
| fog | 0.8997 | 0.7194 | 0.7263 | 0.8191 | 0.8811 | **0.9463** | 0.9061 |
| Gaussian blur | 0.7273 | 0.7907 | 0.8213 | **0.8929** | 0.6453 | 0.8840 | 0.9448 |
| glass blur | 0.5677 | 0.8046 | 0.8017 | **0.8770** | 0.4735 | 0.7621 | 0.8503 |
| impulse noise | 0.5428 | 0.8308 | 0.6881 | 0.5999 | 0.3619 | **0.8560** | 0.9016 |
| jpeg compression | 0.8009 | **0.9078** | 0.8541 | 0.8405 | 0.6395 | 0.8142 | 0.8807 |
| motion blur | 0.8079 | 0.7715 | 0.8045 | **0.8605** | 0.7206 | 0.8491 | N/A |
| pixelate | 0.7317 | 0.8983 | 0.8531 | **0.9156** | 0.6234 | 0.7066 | 0.9369 |
| shot noise | 0.6773 | **0.9233** | 0.8275 | 0.7447 | 0.5374 | 0.7834 | 0.9342 |
| snow | 0.8505 | 0.8835 | 0.8258 | 0.8688 | 0.7929 | **0.8939** | N/A |
| speckle noise | 0.7041 | **0.9171** | 0.8183 | 0.7502 | 0.5603 | 0.8125 | 0.9352 |
| zoom blur | 0.8046 | 0.8163 | 0.8279 | 0.8987 | 0.6514 | **0.8994** | 0.9412 |
| average | 0.7728 | 0.8292 | 0.8095 | 0.8299 | 0.6754 | **0.8613** | N/A |
| mCE | 1.000 | 0.9831 | 1.0825 | 0.8924 | 1.4449 | **0.6376** | N/A |

**Table 3:** Test accuracy on clean images and all the 15 corruptions in CIFAR-10-C. We compare 6 models: the naturally trained model, Gaussian data augmentation with parameter $0.1$, adversarially trained model, low pass filter front end with bandwidth 15, high pass filter front end with bandwidth 31, and AutoAugment without brightness and contrast. Every test accuracy for the corruptions is obtained by averaging over 5 severities. The "average" row provides the average test accuracy over all the corruptions. We also present the results for the all-but-one training. More specifically, for a given corruption type and severity, we train on all the other corruptions at the same severity and evaluate on the given one. Due to some software dependency issue, we were not able to implement two of the corruptions on the training data, therefore, we only report the all-but-one results for 13 of the 15 corruptions. The test accuracy on clean images of all-but-one is averaged over all the "all-but-one" models. Since there test accuracies are not achieved by a single model, we do not compare them with other models, nor do we calculate the average corruption test accuracy and mCE.

## B Fourier heat maps

In this section, we provide the Fourier heat maps for the intermediate layers of the model. We first define the Fourier heat map of the output of a layer. Recall that the $H$-layer feedforward neural network is a function that maps $X$ to a vector $z \in \mathbb{R}^K$, known as the logits. Let $W_h$ be the weights and $\rho_h$ be the possibly nonlinear activation in the $h$-th layer. We let

$$z_h(X) = \rho_h(\cdots \rho_2(\rho_1(X, W_1), W_2) \cdots, W_h) \in \mathbb{R}^{p_h}$$

be the output of the $h$-th layer and thus the logits $z(X) = z_H(X)$. The model makes prediction by choosing $y = \arg\max_k z(X)[k]$. Recall that for a validation image $X$, we can generate a perturbed

|  | natural | Gauss | low pass | high pass | AutoAugment |
|---|---|---|---|---|---|
| clean images | 0.7623 | 0.7425 | 0.7082 | 0.7500 | **0.7725** |
| brightness | 0.6975 | 0.6687 | 0.6214 | 0.6923 | **0.7406** |
| contrast | 0.4449 | 0.3578 | 0.3473 | 0.4911 | **0.5656** |
| defocus blur | 0.5023 | 0.5294 | **0.5803** | 0.4414 | 0.5414 |
| elastic transform | 0.5637 | 0.6000 | **0.6211** | 0.5255 | 0.5846 |
| fog | 0.5715 | 0.4736 | 0.4031 | 0.6459 | **0.6534** |
| frosted glass blur | 0.4187 | 0.5217 | **0.6000** | 0.3460 | 0.5073 |
| Gaussian noise | 0.4492 | **0.6956** | 0.4897 | 0.3979 | 0.5798 |
| impulse noise | 0.4210 | **0.6785** | 0.4736 | 0.3737 | 0.5832 |
| jpeg compression | 0.6630 | **0.6997** | 0.5688 | 0.6388 | 0.6893 |
| pixelate | 0.5826 | 0.6173 | 0.6790 | 0.5237 | **0.6814** |
| shot noise | 0.4294 | **0.6820** | 0.4894 | 0.3837 | 0.5845 |
| zoom blur | 0.3663 | 0.3653 | **0.4177** | 0.2826 | 0.3398 |
| average | 0.5092 | 0.5741 | 0.5243 | 0.4785 | **0.5876** |

**Table 4:** Test accuracy on clean images and 12 corruptions in ImageNet-C. Instead of using the compressed ImageNet-C images provided in [17], the models are evaluated on the corruptions applied in memory. Due to some software dependency issue, we were not able to implement 3 of the 15 corruptions in memory, and thus we only the report test accuracy for 12 corruptions. We compare 5 models: the naturally trained model, Gaussian data augmentation with parameter 0.4, low pass filter front end with bandwidth 45, high pass filter front end with bandwidth 223, and AutoAugmentation. Every test accuracy for the corruptions is obtained by averaging over 5 severities.

image with Fourier basis noise, i.e., $\widetilde{X}_{i,j} = X + rvU_{i,j}$. We then compute layers' outputs $z_h(X)$ and $z_h(\widetilde{X}_{i,j})$, given the clean and perturbed images, respectively, and obtain $\|z_h(X) - z_h(\widetilde{X}_{i,j})\|$ as the model's output change at the $h$-th layer. We conduct this procedure for $n$ validation images $X^{(1)}, \ldots, X^{(n)}$, compute the average output change, and use this average as a measure of the model's stability to the Fourier basis noise. More specifically, we generate the Fourier heat map of the $h$-th layer, denoted by $Z_h \in \mathbb{R}^{d_1 \times d_2}$, as a matrix with entries $Z_h[i,j] = \frac{1}{n} \sum_{\ell=1}^{n} \|z_h(X^{(\ell)}) - z_h(\widetilde{X}_{i,j}^{(\ell)})\|$.

In Figure 8, for 5 different models, we demonstrate the Fourier heat maps for the outputs of 5 layer outputs in the WideResNet architecture: the output of the initial convolutional layer, the outputs of the first, second, and third residual block, and the logits, and we also provide the test error heat map in the last column. In Figure 9, we plot the test error Fourier heat map for two ImageNet models.

## C  Experiment detail

In Figure 1, we visualize the high pass filtered images using normalization. The specific method is as follows. For an image $X \in [0,1]^{d_1 \times d_2}$ (for RGB images we can divide the pixel values by 255), we compute the mean and standard deviation of all the pixels:

$$\bar{X} = \frac{1}{d_1 d_2} \sum_{i,j} X_{i,j}$$

$$s_X = \left( \frac{1}{d_1 d_2} \sum_{i,j} (X_{i,j} - \bar{X})^2, \right)^{1/2}$$

and then the normalized image is defined as

$$X_{\text{norm}} = \frac{1}{s_X} (X - \bar{X}).$$

In Figure 1, we visualize $X_{\text{norm}}$ using the imshow function in the matplotlib.pyplot python package.

| | init conv | 1st res block | 2nd res block | 3rd res block | logits | test error |
|---|---|---|---|---|---|---|

**Figure 8:** Model heat maps for naturally trained model, Gaussian data augmentation, adversarially trained model, data augmentation with "fog noise" at severity 3 (additive noise that matches the Fourier statistics of fog-3 corruption), and AutoAugment.

**Figure 9:** Fourier heat map of ImageNet models with perturbation $\ell_2$ norm 40. In a large area around the center of the Fourier spectrum, the model has test error at least 95%. First row: heat map of the full Fourier spectrum ($224 \times 224$); second row: heat map of the $63 \times 63$ low frequency centered square in the Fourier spectrum.