[Reviews · NeurIPS 2019]

Reviewer 1



The paper is well-written and offers a plethora of detailed information to support its claims, including: -- The interpretation of Gaussian data augmentation as inducing low=pass filtering by the network. This observation, while not entirely surprising, in made within a principled framework for evaluating augmentation techniques. --An investigation of the effectiveness of the AutoAugment approach for data-driven augmentation strategies, pointing at the effectiveness of more diversified augmentation strategies. -- The observation that adversarial attacks are still possible, even after low-pass filtering. and suggesting connection with the DeepViz approach for visualizing the inner workings of CNNs. Overall the results of the paper present work that feels preliminary, in that they consist mostly of a set of insightful observations and do not, at this point, present a coherent and principled framework of addressing data augmentation strategies for ensuring robustness in vision algorithms. However, within its limited scope, the paper is quite thorough and it is very likely to lead others to further investigate these fundamental and important issues. This reviewer has read the authors' rebuttal. After the ensuing discussion, the final score assigned remains unchanged.

Reviewer 2



This paper explores a useful direction to study adversarial robustness and some experimental results add value to understanding this topic. However, some ad hoc choices and some concerns about the experimental evaluation in this empirical study detrimentally impacts the quality of this paper. More details below. Originality: The work seems original to me. Clarity: The presentation is mostly clear. - (line 18) What are i.i.d. benchmarks? Conversely, what are non-i.i.d ones? Do you mean training and test distributions are the same? Quality/ Technical Correctness and Experimental Evaluation (clubbed together as this is an empirical study): The framing of the problem and technical (design) choices need to be clarified. - The main premise of the paper is a bit hazy. Since adversarial training is being used for robustness but testing is not against adversarial attacks/ perturbations but predefined ‘parametric’ ones (brightness, contrast etc. as mentioned in Figure 2), is the goal to be robust against ‘natural’ adversaries, i.e. those caused by natural distribution shifts which occur, when say, an ML system is deployed? Are malicious adversaries considered? If only the former, this point should be carefully elaborated and studied. Perhaps this is what ‘common corruptions’ alludes to (lines 27-30, 126-127). - Another aspect that needs clarification is the reason for using the Fourier domain (lines 35-36, 112-116) for the study. The domain assumes (spatial) stationarity and is used for modeling and studying the properties of linear, time invariant systems. - The observation that in adversarial robustness is being achieved in a part of the Fourier space and not everywhere doesn’t just suggest that more diverse set of augmentations are needed (lines 11-12, 40-43), but that a strategy for choosing this diverse set may be defined in the Fourier space but this is never investigated. - What is the impact of changing the strength of pure harmonic perturbations (v in line 78)? - The impact of varying the bandwidth of the bandpass signal is not at all clear and discussed properly (Figure 5, lines 147 – lines 173). Should the impact of increasing bandwidth be similar to that from smoothing the responses one observes in Figure 3 since AWGN has a flat spectrum and BPF is akin to adding all frequencies in the band in equal amounts? It’s not at all clear what clarity does using a variety of bandwidth choices brings to understanding the phenomenon the paper is trying to study. Similarly, the different choices of the magnitude of added noise (l_2 – lines 147, l_2 = 4, 15.7 in Fig 3 and 4 respectively) is not well motivated. - Section 4.2 (lines 174-189) needs more clarity. It’s not clear if the Fog noise model is a good model for fog corruption. Is fog corruption expected to be an LTI filter? If so, then it raises doubts about the results in Table 1. If not, then the assumptions for the experiment is Sec 4.2 are not met. But this can be studied by itself instead of the speculative musings in lines 185 – 189. - The motivation for Section 4.3 to use a more varied set of augmentations – this is reasonable but not informed (non-trivially) by the study. In other words, it’s a trivial statement to make. The two known augmentation strategies – AutoAugment and SIN+IN used the study are not really suggested by the previous sections. One would expect to propose an augmentation strategy in the Fourier domain to achieve robustness across the entire Fourier spectrum. - In the experiment to study adversarial training (Section 4.4, Figure 7), Madri’s PGD attack is used to generate adversarial examples. The set of adversarial samples generated is a function of the attack. It’s not clear if generic statements can be drawn about adversarial training based on a single adversarial attack model. Secondly, the comparison between the statistics of natural images and the adversarial perturbations seems misplaced. It’s clear at all what the spectrum of the entire gamut of natural images has to do with the spectrum of adversarial perturbations. The latter is more a statement on the classification boundaries of the trained model.

Reviewer 3



Originality: The frequency-domain interpretation of the effects of noise in training data as well as adversarial noise is original. . Quality: By analyzing common corruptions and model performance in the frequency domain, the paper establishes connections between frequency of a corruption and model performance under data augmentation. Clarity: The paper is easy to follow. Significance:: The analysis is made firm with results on a benchmark data.

[Author Response · NeurIPS 2019]

Thank you to the reviewers for their comments, we are happy that all reviewers agreed that the Fourier analysis we introduce provides novel and useful insights towards understanding model robustness. We will focus this response on addressing concerns raised by Reviewer 2.

**R2: "Are malicious adversaries considered?"**

As stated in our abstract and introduction, our work is centered on the problem of distribution shift and not specifically adversarial robustness. We are primarily interested in understanding how a low (or high) frequency bias of image models affects robustness to low (or high) frequency corruptions. We consider the primary contribution of our work to be the Fourier analysis that we introduce, which we believe will prove useful for future research on robustness.

**R2: "Section 4.2 needs more clarity. It's not clear if the Fog noise model is a good model for fog corruption..."**

The purpose of this section was to demonstrate an asymmetry that occurs when augmenting in the low vs high frequency domain. Figures [3,4,5,6] all demonstrate that Gaussian data augmentation biases the model towards low frequency statistics in the image. This results generally in improved robustness to high frequency corruptions, while reducing robustness to low frequency corruptions. A natural question is what happens if we augment with low frequency noise, do we then bias the model towards high frequency statistics and improve robustness to low frequency corruptions? This appears not to hold, as demonstrated by the fog noise experiments. We will rework this section to be more clear as to the motivation for the considered experiment.

Note the goal is not to specifically design a data augmentation that is a good model for fog. Our problem setting is distribution shift, here fog is used as an example of a domain shift that is unknown at training time. Performing well on fog is trivial if the model is trained specifically on fog (e.g. see Figure 4 in another related work *MNIST-C: A Robustness Benchmark for Computer Vision*).

**R2: "... a strategy for choosing this diverse set may be defined in the Fourier space but this is never investigated"**

The experiments in Section 4.2 are specifically exploring data augmentation in the Fourier space. However, because augmenting on low frequency Fourier noise does not transfer to other low frequency corruptions, we decided to investigate more sophisticated augmentation strategies (e.g. AutoAugment). Note as well that Gaussian data augmentation is mathematically equivalent to adding all Fourier basis vectors with i.i.d Gaussian coefficients.

**R2: "The motivation for Section 4.3 to use a more varied set of augmentations – this is reasonable but not informed (non-trivially) by the study."**

A large motivation for exploring AutoAugment was specifically the experiments in 4.2. Furthermore, the fact that we discovered that the method achieves SOTA on Imagenet-C is certainly relevant to our work and worth including. Additionally, we applied our Fourier analysis to this model and discovered that while it appears to improve robustness on the Imagenet-C corruptions, there exist Fourier frequencies for which performance is degraded relative to the naturally trained model (see the high frequencies in Figure 4). This demonstrates that the Fourier analysis we introduce can identify blind spots in models for which the Imagenet-C benchmark misses. We believe this discovery can lead to additional corruptions to add to this benchmark in order to achieve a more complete perspective on model robustness.

**R2: "It's not clear if generic statements can be drawn about adversarial training based on a single adversarial attack model."**

We believe there was some confusion over what we were arguing in section 4.4. Reviewer 2 is absolutely correct that one should be careful about drawing conclusions about adversarial examples by analyzing a single attack, this in fact is exactly what we were discussing in paragraph 3. As discussed, the statistics of adversarial examples will ultimately depend on how the adversary generates the perturbation. We will update this section to make it more clear that what Figure 7 is demonstrating is that the statistics of the model gradients (as measured by applying a few steps of PGD) confirm our overall hypothesis about how adversarial training biases the model towards lower frequency statistics in the image. Note, the experiments in Figures [3,5,6] also confirm this hypothesis. Taken together, the experiments in the whole paper lend support to the statement that adversarial training biases the model towards low frequency statistics in the data. We will be more clear that it would be incorrect to conclude that all possible adversarial perturbations must have these properties. In fact, Figure 3 demonstrates that errors exist in the worst-case at all frequencies.

**R3: "Asking for more experiments than just CIFAR-10"**

Please note that the AutoAugment results in Table 2 are on Imagenet-C. Additionally, the Fourier analysis in Figure 4 is for models trained on Imagenet. We are also updating the paper to include new experiments on Imagenet akin to Figure 1, where the models are trained and tested with extreme high and low pass filtering applied. What we find on Imagenet is interesting, models can achieve well over 50% accuracy using only high frequency features which are typically invisible to the human eye. We also demonstrate a rescaling technique that can be used to visualize these features.

[Meta-Review · NeurIPS 2019]

The paper proposes an interesting angle to investigate robustness of convnets, looking at Fourier spectrum of images and/or perturbations. Two of the reviewers were very positive, while R2 raised some concerns and ultimately was not completely satisfied by the rebuttal. However, in discussion among the reviewers and the AC, everybody agreed that the paper has potentially important contributions. Despite the shortcomings I recommend to accept the paper as a poster. I do recommend that the authors aim to take the detailed comments and improvement suggestions by all the reviewers into account. [This meta-review was reviewed and revised by the Program Chairs]